# Amazonian Bacteria from River Sediments as a Biocontrol Solution against *Ralstonia solanacearum*

**DOI:** 10.3390/microorganisms12071364

**Published:** 2024-07-03

**Authors:** Jennifer Salgado da Fonseca, Thiago Fernandes Sousa, Suene Vanessa Reis de Almeida, Carina Nascimento Silva, Gleucinei dos Santos Castro, Michel Eduardo Beleza Yamagishi, Hector Henrique Ferreira Koolen, Rogério Eiji Hanada, Gilvan Ferreira da Silva

**Affiliations:** 1Graduate Program in Biotechnology, Federal University of Amazonas, Manaus 69080-005, AM, Brazil; fonseca.jsd@gmail.com (J.S.d.F.); thiago-fernandes2@hotmail.com (T.F.S.); 2Graduate Program in Agriculture in the Humid Tropics, National Amazon Research Institute, Manaus 69060-062, AM, Brazil; suenevanessa5@gmail.com (S.V.R.d.A.); cncarinaczs@gmail.com (C.N.S.); rhanada.inpa@gmail.com (R.E.H.); 3Graduate Program in Biodiversity and Biotechnology, State University of Amazonas, Manaus 69065-001, AM, Brazil; gleucineii@gmail.com (G.d.S.C.); hkoolen@uea.edu.br (H.H.F.K.); 4Embrapa Agricultura Digital, Campinas 13083-970, SP, Brazil; michel.yamagishi@embrapa.br; 5Embrapa Amazônia Ocidental, Manaus 69010-970, AM, Brazil

**Keywords:** actinomycetes, phylogenomic identification, dDDH, ANI, bioactive metabolites, bioprospecting

## Abstract

Bacterial wilt, caused by *Ralstonia solanacearum*, is one of the main challenges for sustainable tomato production in the Amazon region. This study evaluated the potential of bacteria isolated from sediments of the Solimões and Negro rivers for the biocontrol of this disease. From 36 bacteria selected through in vitro antibiosis, three promising isolates were identified: *Priestia aryabhattai* RN 11, *Streptomyces* sp. RN 24, and *Kitasatospora* sp. SOL 195, which inhibited the growth of the phytopathogen by 100%, 87.62%, and 100%, respectively. These isolates also demonstrated the ability to produce extracellular enzymes and plant growth-promoting compounds, such as indole-3-acetic acid (IAA), siderophore, and ammonia. In plant assays, during both dry and rainy seasons, *P. aryabhattai* RN 11 reduced disease incidence by 40% and 90%, respectively, while promoting the growth of infected plants. *Streptomyces* sp. RN 24 and *Kitasatospora* sp. SOL 195 exhibited high survival rates (85–90%) and pathogen suppression in the soil (>90%), demonstrating their potential as biocontrol agents. This study highlights the potential of Amazonian bacteria as biocontrol agents against bacterial wilt, contributing to the development of sustainable management strategies for this important disease.

## 1. Introduction

Bacterial wilt, caused by the *Ralstonia solanacearum* species complex (RSSC), is one of the most devastating diseases affecting tomato production worldwide [1,2,3]. In Brazil, losses due to this disease are particularly significant in the northern region, such as in the state of Amazonas, where losses of 40–80% are reported in the production of vegetables, including tomato, bell pepper, scarlet eggplant, chili pepper, and eggplant, as well as in banana (*Musa paradisiaca*) cultivation. In addition to solanaceous crops and banana, beach daisy (*Melanthera discoidea*) and miracle tree (*Moringa oleifera*) have also been reported as hosts in the state [4,5,6,7].

The RSSC, previously classified into phylotypes (I, II, III, IV), has recently been reclassified into three species: *R. solanacearum* (phylotype II), *R. pseudosolanacearum* (phylotypes I and III), and *R. syzygii* (phylotype IV) [8]. Studies suggest Brazil as a possible center of diversity for phylotype II, although phylotype I strains are also found in the country [7,9,10,11,12,13,14,15].

Controlling bacterial wilt is challenging due to the versatile lifestyle of *R. solanacearum,* which allows its adaptation to different ecological niches, such as soil, water, and plants (non-host plant rhizosphere and host xylem), and the ability to survive in the soil for long periods [10,16,17,18]. The pathogen infects plants through the roots, invades the xylem, and spreads throughout the aerial portion via the vascular system, multiplying intensely and producing exopolysaccharides (EPS) that obstruct the vessels, causing wilting symptoms and, eventually plant death [19,20].

Current strategies for controlling bacterial wilt in tomato include the genetic improvement of resistant cultivars, such as the Yoshimatsu cultivar, developed to adapt to the climatic conditions of the Amazon region [21,22,23]. However, this cultivar still faces challenges in consumer acceptance due to characteristics such as fruit size and cracking when ripe. Furthermore, the use of chemicals like validamycin A and validoxylamine to induce plant resistance is no longer recommended for application in tomato plants [24,25,26]. Although a mixture of lipopeptides produced by *Bacillus amyloliquefaciens* has shown potential for controlling the phytopathogen, no options are currently available on the market [27].

In this context, the formulation of biodefensives using bacteria capable of suppressing the phytopathogen through multiple mechanisms, such as competition, resistance induction, and the production of antibiotics, siderophores, and/or cell wall-degrading enzymes, has emerged as a promising approach for promoting sustainable agriculture [28,29,30,31,32,33,34,35,36,37,38]. Therefore, the objective of this study was to explore Amazonian microbial genetic resources, specifically bacteria isolated from sediments of the Negro and Solimões rivers, as a potential source for the biocontrol of *R. solanacearum*, focusing on the development of environmentally sustainable strategies for the control of bacterial wilt in tomato plants.

## 2. Materials and Methods

### 2.1. Bacterial Isolates

Thirty-six bacterial isolates obtained from sediments of the Solimões and Negro Rivers were used, which are preserved in the Laboratory of Genomics and Applied Microbiology of the Legal Amazon (GENAGRO) at Embrapa—CPAA. The strains were maintained in LB media for non-filamentous bacteria, incubated for 24 h at 28 °C, and ISP2 for actinobacteria, incubated for 7 days at 28 °C. Access to the genetic heritage was authorized by SISGEN N^o^. A39C76B.

### 2.2. In Vitro Antimicrobial Activity against Ralstonia Solanacearum

The antagonism of 36 bacterial isolates from the Negro and Solimões rivers (Appendix A) against *R. solanacearum* was evaluated in vitro using paired culture tests adapted from Velho-Pereira and Kamat [36]. In Petri dishes containing LB medium (non-filamentous bacteria) and ISP2 (actinobacteria), a 7 cm vertical streak of the antagonists was made and incubated for 48 h for LB plates and seven days for ISP2 plates at 28 °C. Subsequently, a 3 cm streak of *R. solanacearum* was made at a distance of 1 cm from the incubated antagonist for 24 h. All assays were performed in triplicate. At the end of the assays, with the aid of a caliper, the length and width of the pathogen were measured to calculate the growth area. The determination of phytopathogen inhibition was performed according to the following formula:$$PASDAAS(\%)=\frac{AWG}{TGA}\times 100$$
where PASDAAS represents the percentage of specific antibiotic activity of the area score, AWG the area without growth, and TGA the total growth area of the pathogen. To calculate AWG, the area present in the treatment plate was subtracted by the TGA of the control plate.

### 2.3. Biocontrol Evaluation under Greenhouse Conditions

In planta evaluation in a greenhouse was performed in 1-L pots containing Vivatto plus^®^ (São Paulo, Brazil) substrate based on a completely randomized design (CRD) with 20 replicates for each treatment using the San Marzano cultivar (Isla^®^, Porto Alegre, Brazil), which is susceptible to bacterial wilt caused by *R. solanacearum*. In the preliminary study, isolates RN 11, RN 24, and SOL 195 were tested in September (32 ± 6.5 °C and humidity of 52 ± 8%), and the final test was performed with the best-performing isolate in December (28 ± 1.5 °C and humidity of 71 ± 10%) 2023. Tomato seedlings grew in a seedbed for 30 days, but on the 23rd day, the seedlings referring to the treatments with the isolates were inoculated with 5 mL of cell suspension (10^10^ CFU mL^−1^) of each isolate, while in the negative and positive controls, 5 mL of distilled water was added. On the 30th day, the seedlings were transplanted, and after 5 days, small incisions were made in the roots at a distance of 2 cm from the collar of all seedlings for infection with 5 mL of *R. solanacearum* suspension (10^10^ CFU mL^−1^) in the positive control and treatments. In the negative control, 5 mL of sterile distilled water was added. The calculation of the percentage of disease incidence is given by
$$Disease\;incidence\left(\%\right)=\frac{DP}{TP}\times 100$$
where DP is the number of diseased plants and TP is the total number of plants in the experiment. Tomato plants were considered to be diseased when they presented at least one of the symptoms: wilted leaves and wilted branches. Symptom monitoring was performed daily for 30 days, and to avoid misinterpretation of the seedlings’ condition, the evaluation was performed 1 h after irrigation. Survival was calculated using the following formula:$$Survivor(\%)=\frac{LP}{TP}\times 100$$
where LP is the number of live plants and TP is the total number of plants in the experiment. To evaluate the effect of the biocontrol agent on aspects related to plant development in the presence of the pathogen, the following were measured: height (cm), stem diameter (cm), root length (cm), as well as shoot dry weight (g) and root dry weight (g). The seedlings were measured using a measuring tape and caliper.

### 2.4. Analysis of R. solanacearum Suppression in Soil

At the end of the *in planta* tests, soil samples were collected from all treatments and controls. In a test tube, 1 g of soil was resuspended in 10 mL of sterile distilled water and shaken. From this tube, serial dilution was performed up to a concentration of 10^−8^. In triplicate, 100 µL of the 10^−1^, 10^−5^, and 10^−8^ concentrations was plated on Petri dishes containing LB medium and incubated at 28 °C for 7 days, with daily monitoring of colony appearance. Bacterial colonies with colorimetry similar to that of the phytopathogen were inoculated on CPG (casamino acid-peptone-glucose) and TTC (triphenyl tetrazolium chloride) media for confirmation, where the formation of opaque white colonies on CPG medium turned dark pink on TTC medium, which was considered indicative of the presence of *R. solanacearum* [37,38]. Suppression was calculated using the following formula:$$Suppression(\%)=100-\left(\frac{n{}^{\circ}\;of\;R.\;solanacearum\;colonies\;in\;treatment}{n{}^{\circ}\;of\;R.\;solanacearum\;colonies\;in\;PC}\times 100\right)$$

### 2.5. DNA Extraction, Sequencing, and Genome Assembly

Only the isolates selected for in planta tests were identified. Isolate RN 11 was cultured in LB medium for 24 h, while isolates RN 24 and SOL 195 were cultured in ISP2 medium for 96 h. The cultures were centrifuged, and the supernatants were discarded to obtain the cell mass. DNA was isolated using the CTAB protocol [39]. The amount of DNA obtained was estimated by spectrophotometry (NanoDrop 2000, Thermo Scientific, Waltham, MA, USA), while integrity was verified by electrophoresis on 0.8% (*w*/*v*) agarose gel. The Illumina platform (150 bp paired-end) was used for complete genome sequencing, with a minimum sequencing coverage of 100X. Genome De Novo assembly was performed using SPAdes assembler [40], kmer = 123, read correction algorithm was also performed in order to reduce the number of mismatches and short indels.

### 2.6. Phylogenomic Identification

The identification of isolates at the species level was performed based on the complete genome through comparison with type species using the TYGS platform (https://tygs.dsmz.de, accessed on 8 January 2024). From the most closely related species identified in TYGS, the dDDH calculation was obtained using the d2 formula with the aid of the GGDC platform (https://ggdc.dsmz.de/ggdc.php#, accessed on 10 January 2024) and the ANI (Average Nucleotide Identity) calculation was performed using the OAT software (https://www.ezbiocloud.net/tools/orthoani, accessed on 10 January 2024), where dDDH < 70% and ANI < 95% were used as indicative of a new species [41,42,43]. For the search of plasmids in the genomes, they were analyzed through Plasmidfinder, available on the Galaxy Europe platform (https://usegalaxy.eu/, accessed on 15 February 2024).

### 2.7. Production of Extracellular Enzymes

For extracellular enzymes, the assays were performed only for the isolates selected for biocontrol tests in specific media for amylase [44], cellulase [45], lipase [46], protease [47], and chitinase [48]. Starch (amylase), skimmed milk (protease), Tween 80 (lipase), chitin (chitinase), and carboxymethyl cellulose (cellulase) were used as substrates in the enzymatic tests. All assays were performed in triplicate with 5 mm discs of bacterial cultures and incubated for 48 h at 28 °C for subsequent measurement of halos (mm) with the aid of a caliper.

### 2.8. Production of In Vitro Growth Promotion Inducers

All the assays described below were performed with the selected isolates (RN 11, RN 24, and SOL 195) for the biocontrol tests in tomato plants.

#### 2.8.1. Phosphate (P) and Zinc (Zn) Solubilization

The P solubilization was performed in Pikovskaya’s medium [49], while Zn solubilization was performed according to Saravanan et al. [50] with modifications, where the medium was composed of 798 mL of distilled water, 200 mL of M95X solution (33.78 g L^−1^ Na_2_HPO_4_, 15 g L^−1^ KH_2_PO_4_, 2.5 g L^−1^ NaCl, and 5 g L^−1^ NH_4_Cl), 2 mL of 1M MgSO_4_.7H_2_O solution, 100 µL of 1 M CaC_2_.2H_2_O solution, 15 g of agar, and 1 g of Zn source. ZnO and ZnSO_4_ were used as Zn sources. All assays were performed in triplicate with 5 mm discs of isolate cultures and incubated for five days at 28 °C for subsequent measurement of halos with the aid of a caliper.

#### 2.8.2. Siderophore

The assay was performed according to the modifications made by Thampi and Bhai [51] to the assay described by Schwyn and Neilands [52], where MGs-1 medium was used (20 g L^−1^ dextrose, 1 g L^−1^ KNO_3_, 0.1 g L^−1^ NaCl, 0.1 g L^−1^ MgSO_4_.7H_2_O, 0.5 g L^−1^ K_2_HPO_4_, 15 g L^−1^ agar, 900 mL of distilled water, and 100 mL of CAS). All assays were performed in triplicate with 5 mm discs of isolate cultures and incubated for five days at 28 °C for subsequent measurement of halos with the aid of a caliper.

#### 2.8.3. Indole Acetic Acid (IAA)

Isolate RN 11 was cultured in LB medium, and isolates RN 24 and SOL 195 were cultured in ISP2 medium, both supplemented with tryptophan (150 mg L^−1^), in triplicate under agitation at 150 rpm in the dark for 7 days. The cultures were centrifuged for 30 min at 4000 rpm to obtain the supernatant. The assay was performed with 1 mL of supernatant and 1 mL of Salkowski’s solution. The reaction was incubated for 60 min in the dark for subsequent reading at 595 nm. In the blank, sterile culture medium was used instead of the supernatant. For quantification, a standard curve with IAA was made [51].

#### 2.8.4. Ammonia

The isolates were cultured in triplicate in peptone-water medium (10 g L^−1^ peptone, 5 g L^−1^ NaCl) for 48 h at 30 °C. The culture was centrifuged for 30 min at 4000 rpm, and 10 mL of the supernatant was transferred to test tubes. Five hundred microliters of Nessler’s solution was added, and the color change to yellow or brown was observed [53].

### 2.9. Statistical Analysis

The data were subjected to analysis of variance (ANOVA) using R 4.3.1. software, and when significant (*p* < 0.05), the Scott-Knott mean test was applied. The assumptions of ANOVA, such as normality and homoscedasticity of residuals, were verified using the Shapiro–Wilk and Levene’s tests, respectively.

## 3. Results

### 3.1. In Vitro Antimicrobial Activity

Of the 36 isolates from sediments of the Solimões and Negro rivers evaluated against *R. solanacearum*, only three (RN 2, SOL 110, and SOL 229) did not show antibiotic activity. Twenty-nine isolates exhibited PASDAAS (percent area specific differential antibiotic activity score) between 7 and 36%, while isolates RN 11, RN 24, SOL 116, and SOL 195 stood out with the highest indices (Appendix A). With the exception of SOL 116 (67% antibiotic activity), the others presented indices equal to or greater than 70% and were selected for biocontrol evaluation in the greenhouse. Isolates RN 11 and SOL 195 completely inhibited the growth of *R. solanacearum*, and RN 24 inhibited 87.62% (Appendix A).

### 3.2. Phylogenomic Identification

Phylogenomic analyses of isolates RN 11, RN 24, and SOL 195 were performed using ANI and dDDH. The results revealed that isolated RN 11 belongs to the species *Priestia aryabhattai*, with ANI and dDDH values of 98.61% and 88.3%, respectively. On the other hand, isolates RN 24 and SOL 195 presented ANI and dDDH values below the cutoff points for new species, and the most closely related type species were *Streptomyces ardesiascus* and *Kitasatospora aureofaciens*, respectively (Table 1). RN 24 exhibited 92.29% ANI and 46.8% dDDH with *S. ardesiascus*, while SOL 195 presented 86.36% ANI and 31.1% dDDH with *K. aureofaciens*.

### 3.3. Production of Growth Promotion Inducers and Enzymes 

The selected isolates were evaluated for the production of extracellular enzymes and plant growth inducers (Table 2). *Kitasatospora* sp. SOL 195 stood out in the production of amylase (20 ± 1.2 mm) and chitinase (18.2 ± 0.9 mm), while *P. aryabhattai* RN 11 showed the highest production of lipase (26 ± 1.5 mm) and was the only one to produce protease (17.7 ± 1.1 mm). *Streptomyces* sp. RN 24 exhibited the highest production of cellulase (31 ± 1.8 mm) (Appendix A).

All isolates produced indole-3-acetic acid (IAA) and ammonia, with the strain *Kitasatospora* sp. SOL 195 presents the highest levels for both compounds and RN 11 the lowest. *Streptomyces* sp. RN 24 did not produce P and Zn solubilizers or siderophores under the tested conditions (Table 2). *P. aryabhattai* RN 11 was the only one to solubilize Zn (both sources) and showed the highest production of siderophores.

### 3.4. Biological Control under Greenhouse Conditions

In the Amazonian dry period (September), plants inoculated with *P. aryabhattai* RN 11 showed symptoms after 10 days, while in plants inoculated only with the pathogen (positive control), symptoms were observed 24 h post-inoculation. Plants inoculated with the microbial agents *Streptomyces* sp. RN 24 and *Kitasatospora* sp. SOL 195 demonstrated symptoms from the third and sixth day, respectively. In this period, when the survival index was evaluated, the positive control presented an index of 65%, while this value increased to 85% in the treatment with SOL 195 and 90% with RN 11 and RN 24. Treatment with RN 11 reduced disease incidence by 40%, followed by RN 24 and SOL 195, which reduced incidence by 20% and 5%, respectively. In addition, all isolates suppressed the pathogen in the soil with indices > 90% (Figure 1).

Still in the dry period, in addition to the control of *R. solanacearum,* isolates RN 11, RN 24, and SOL 195 were also evaluated for their positive effect on parameters related to growth promotion under conditions of infection with *R. solanacearum* (Table 3). The results obtained demonstrated that treatment with RN 24 provided a significant increase in plant height compared to the positive control (PC), reaching values similar to the negative control (NC). Regarding stem diameter and leaf size, all treatments, except for those still in the dry period, in addition to the control of *R. solanacearum*, isolates RN 11, RN 24, and SOL 195 were also evaluated for their positive effect on parameters related to growth promotion under conditions of infection with *R. solanacearum* (Table 3). The results obtained demonstrated that treatment with RN 24 provided a significant increase in plant height compared to the positive control (PC), reaching values similar to the negative control (NC). Regarding stem diameter and leaf size, all treatments, except for SOL 195, were significantly superior to PC, with emphasis on RN 11, which did not differ statistically from NC for these parameters. Regarding root growth, all treatments showed significant differences compared to PC, with RN 11 being equal to NC. In addition, all treatments with biocontrol agents resulted in a higher number of branches when compared to PC. Isolates RN 11 and RN 24 stood out in the parameters of shoot dry weight (ADW) and root dry weight (RDW), presenting values significantly higher than PC, although they did not differ from each other.

Based on the best performance obtained in the dry period, the strain *P. aryabhattai* RN 11 stood out from the other isolates based on disease incidence and survival of infected plants, being selected for evaluation in the rainy period. The disease incidence in the positive control in the rainy period was 80% with a mortality rate of 65%. Compared to the previous period, there was a 20% reduction in incidence and a 10% increase in mortality. The same is observed in the treatment with RN 11, which presents a high incidence (60%) in the dry period and low in the rainy period (10%); however, both the mortality rate (10%) and pathogen suppression (>90%) in the soil are similar in both periods (Appendix A).

In the rainy period, the growth promotion parameters are significantly influenced by the treatment with RN 11 compared to the positive control (PC) in all aspects evaluated (Table 4). The height of plants treated with RN 11 (42.04 cm) was significantly higher than both PC (34.19 ± 2.5 cm) and negative control (NC) (37.69 ± 3.1 cm). Stem diameter was also positively affected by RN 11 (0.38 ± 0.03 cm), being statistically superior to PC (0.22 ± 0.02 cm) and NC (0.3 ± 0.09 cm). Regarding root length, treatment with RN 11 (29.04 ± 3.6 cm) did not differ significantly from NC (27 ± 4.3 cm), but both were superior to PC (19.73 ± 3.6 cm). The number of branches was higher in plants treated with RN 11 (8 units) compared to PC and NC (both with 7 units). Leaf size did not differ between RN 11 (5.77 ± 0.5 cm) and NC (5.71 ± 0.4 cm), both being significantly superior to PC (4.77 ± 0.9 cm). Shoot dry weight (ADW) was significantly higher in the treatment with RN 11 (1.32 ± 0.23 g) compared to PC (0.94 ± 0.11 g) and NC (1.1 ± 0.28 g). Root dry weight (RDW) was statistically similar between RN 11 (0.26 ± 0.07 g) and NC (0.266 ± 0.08 g), both being superior to PC (0.126 ± 0.03 g). These results indicate that *P. aryabhattai* RN 11 can act not only as an efficient biocontrol agent but also as a growth promoter in tomato plants grown in the Amazon during the rainy period, significantly improving various plant development parameters compared to the positive control infected with *R. solanacearum*.

## 4. Discussion

The dynamics of rivers and their tributaries promote the exchange of organic matter and microbial agents capable of producing diverse secondary metabolites, creating a unique dynamic in these ecosystems that favors the emergence and diversification of microbial lineages, enhancing the discovery of new biological solutions for challenges in health, agriculture, and industry [54,55,56,57,58,59,60]. In this context, Amazonian rivers have proven to be a rich source of microbial biodiversity with the capacity to produce new antimicrobial agents with biotechnological potential, filling the gaps in current knowledge about the diversity and potential of these microorganisms [61,62,63].

Exploring the microbial diversity of two Amazonian rivers with distinct characteristics, the Negro River and the Solimões River, provides a comprehensive view of the biotechnological potential of aquatic microorganisms in the region. The Negro River is considered the largest blackwater river in the world and is characterized by its high acidity (pH < 5.0), high concentration of humic compounds, low sediment load (clay), and low concentration of chemical elements (mainly cations), which is why it has low electrical conductivity [64,65,66]. In contrast, the Solimões River is classified as a whitewater river, with a pH of 5–7, rich in Ca^2+^ and HCO_3_, and a high amount of suspended material and dissolved salts, resulting in a greater diversity of microorganisms [67].

The microbial diversity of the Amazonian aquatic environment, exemplified in this study by the exploration of microorganisms from the sediments of the Solimões and Negro rivers, reveals that this ecosystem can be an important source for the development of new inputs for the biological control of *R. solanacearum*, a serious problem for agriculture, especially related to vegetable production in northern Brazil. The results obtained in this study fill a gap in knowledge about the potential of Amazonian aquatic microorganisms for controlling this economically important phytopathogen.

The antibiotic activity identified against *R. solanacearum*, based on the Percent Area Specific Differential Antibiotic Activity Score (PASDAAS), showed variation among the selected isolates RN11, RN 24, and SOL195, with inhibition ranging from 87.55 to 100%. This variation in antimicrobial efficacy reflects what has been observed in previous studies documenting the metabolic diversity of aquatic microorganisms and their ability to produce bioactive compounds [55,59,68,69,70]. The observed differences in the antibiotic activity of the isolates may be related to the diversity of secondary metabolites produced by each strain and the cultivation conditions used in the assays. Additional studies are needed to elucidate the specific compounds responsible for antibiotic activity and to optimize the production conditions of these metabolites.

The results obtained confirm the identification of RN 11 as *P. aryabhattai,* because the ANI and dDDH values were above the cutoff points. On the other hand, the ANI and dDDH values below the cutoff points for RN 24 and SOL 195 provide strong evidence that these isolates represent new species within the genera *Streptomyces* and *Kitasatospora*, respectively. The ANI and dDDH metrics have been widely employed to delimit bacterial species, offering a robust and reliable alternative to conventional methods. The established cutoff points for species delimitation correspond to 95–96% for ANI and 70% for dDDH [41,42,43]. The discovery of new species of *Streptomyces* and *Kitasatospora* from Amazonian river sediments highlights the importance of this ecosystem as a source of unexplored microbial diversity. Future studies may investigate the biotechnological potential of these new species and their distribution and ecological role in the aquatic environments of the region.

Actinobacteria, such as those of the genera *Streptomyces* and *Kitasatospora*, are known for their range of molecules with antimicrobial, antitumor, and immunosuppressive properties, with emphasis on aspects related to biocontrol and growth promotion in agriculture [35,71,72,73,74,75,76,77,78,79,80,81]. The results of this study provide promising perspectives for the bioprospecting of new secondary metabolites from the RN 24 and SOL 195 strains, contributing to the expansion of the diversity of known bioactive compounds and to the advancement in the discovery of molecules with biotechnological applications.

In addition to secondary metabolites, the production of extracellular enzymes and plant growth inducers by bacteria is an important mechanism for promoting plant growth [82]. Extracellular enzymes, such as amylases, cellulases, chitinase, lipases, and proteases, play fundamental roles in the degradation of complex polymers, making nutrients available to plants and contributing to the suppression of phytopathogens [83]. In this study, the isolates showed different enzymatic production profiles, with emphasis on *Kitasatospora* sp. SOL 195 in the production of amylase and chitinase, *P. aryabhattai* RN 11 in the production of lipase and protease, and *Streptomyces* sp. RN 24 in the production of cellulase. This functional diversity can be exploited for the development of microbial inoculants with multiple enzymatic activities, aiming at promoting plant growth and protection against phytopathogens.

The production of protease by *Bacillus subtilis* B315 was used as evidence of antagonistic potential against *R. solanacearum*, as the enzyme assists in resistance to the phytopathogen by degrading the extracellular polymeric substances (EPS) and the biofilm produced by the pathogen [84]. Considering that *Bacillus* and *Priestia* are closely related genera, there is a possibility that *P. aryabhattai* RN 11 uses this mechanism, differentiating itself from *Streptomyces* sp. RN 24 and *Kitasatospora* sp. SOL 195, which did not demonstrate protease production. Additional studies are needed to elucidate the specific role of the protease produced by *P. aryabhattai* RN 11 in the suppression of *R. solanacearum* and to investigate other antagonism mechanisms that may be involved.

In addition to enzymes, the isolates also produced plant growth inducers, such as IAA, ammonia, and siderophores. IAA is an important plant hormone involved in regulating plant growth and development, while ammonia contributes to nitrogen nutrition [85]. Siderophores, in turn, are iron-chelating compounds that facilitate the absorption of this micronutrient by plants, especially in soils with low iron availability [86]. The production of these plant growth inducers by the isolates suggests their potential for promoting plant growth, in addition to their biocontrol activity against *R. solanacearum.* Future studies may evaluate the effect of these isolates on the growth and development of different agricultural crops, as well as investigate the molecular mechanisms involved in the plant-microorganism interaction.

The ability to solubilize trace nutrients, such as P and Zn, is another relevant mechanism for promoting plant growth. Although none of the isolates solubilized P under the tested conditions, RN 11 stood out in the solubilization of Zn from different sources (ZnO and ZnSO_4_). The solubilization of Zn by actinomycetes can increase the availability of this micronutrient for plants, contributing to their growth and productivity [87]. The ability of *P. aryabhattai* RN 11 to solubilize Zn suggests its potential for application as a biofertilizer, especially in soils deficient in this micronutrient. Additional studies are needed to evaluate the effectiveness of RN 11 in promoting plant growth under field conditions and to investigate the mechanisms involved in Zn solubilization.

The strains RN 11, RN 24, and SOL 195, when evaluated for the biological control of *R. solanacearum* under greenhouse conditions, demonstrated efficacy in suppressing the phytopathogen in the soil > 90%, but only RN 11 showed a high survival rate associated with reduced disease incidence and a significant growth promotion effect compared to the positive control. These results highlight the potential of *P. aryabhattai* RN 11 as an effective biocontrol agent against *R. solanacearum*, capable of suppressing the pathogen in the soil, reducing disease incidence, and promoting plant growth. Future studies may investigate the efficacy of RN 11 under field conditions, as well as evaluate its compatibility with other integrated disease management practices.

Agricultural production losses caused by *R. solanacearum* vary due to several factors such as the level of resistance of the cultivar used, climate, soil, and genetic variation of the *R. solanacearum* strains present in the crop [26]. In this complex context of interactions influenced by various factors, we evaluated the biological control of *R. solanacearum* in the two main climatic seasons that occur in the Amazon region (summer and rainy period). As observed in the obtained results, the isolate RN 11 showed a 40% reduction in the incidence of tomato disease in summer and 90% in the rainy season, exemplifying what was exposed by Yuliar et al. [26], but it is important to note that the climatic effect did not interfere with the suppression of the phytopathogen in the soil or with the survival rate promoted by biological control.

The analysis of the results under different climatic conditions provides insight into the robustness of *P. aryabhattai* RN 11 as a biocontrol agent, demonstrating its efficacy in different environmental contexts. This characteristic is highly desirable for the implementation of large-scale biocontrol strategies, as climatic conditions can vary significantly between different regions and times of the year.

The results obtained with *P. aryabhattai* RN 11 are superior to those observed with the use of improved mutants of *Bacillus amyloliquefaciens* for the biocontrol of *R. solanacearum*. In the study by Yadav et al. [88], conducted under climatic conditions similar to those of the rainy period in the present work, the mutants of *B. amyloliquefaciens* DSBA-11 (MNTG-21, MUV-19, and MHNO2-20) provided a 50–73% reduction in disease incidence and 60–88% survival. These values are lower than those obtained with the RN 11 strain, demonstrating its greater potential in controlling *R. solanacearum*, even when compared to a species already consolidated as a biological control agent for various phytopathogens, such as *B. amyloliquefaciens* [31,89,90,91,92,93,94,95].

The fact that *P. aryabhattai* RN 11 presents superior results to improved mutants of *B. amyloliquefaciens* further highlights the potential of this strain, as it has not undergone genetic modifications to enhance its biocontrol efficiency. These findings open up promising perspectives for the development of microbial inoculants based on *P. aryabhattai* RN 11, with the potential to surpass the efficacy of products already on the market.

*Priestia aryabhattai*, previously known as *Bacillus aryabhattai* [96], was isolated as an endophytic bacterium and has evidence in the literature regarding its potential for the biocontrol of phytopathogens such as *Fusarium oxysporum* [97] and *Ralstonia syzygii* [98], a species that is part of the *R. solanacearum* species complex that causes bacterial wilt [8].

The results of the present study corroborate and expand the knowledge about the potential of *P. aryabhattai* as a biocontrol agent, demonstrating its efficacy against *R. solanacearum* under different climatic conditions and highlighting its potential for application in the integrated management of bacterial wilt in tomato. Furthermore, this is the first report of a *P. aryabhattai* strain isolated from Amazonian river sediments, revealing the importance of this ecosystem as a source of microorganisms with biotechnological potential.

As promising perspectives, these studies open up a range of opportunities for future research aimed at exploring the biotechnological potential of bacteria isolated from Amazonian river sediments. Among the most relevant possibilities is the development of microbial inoculants based on *P. aryabhattai* RN 11 for the integrated management of bacterial wilt in tomato. To this end, additional studies are needed to evaluate the efficacy of this isolate under field conditions, as well as to optimize the formulations and application methods of the inoculant.

Another promising line of research is the investigation of the molecular mechanisms involved in the interaction between *P. aryabhattai* RN 11 and the host plant, as well as in the suppression of *R. solanacearum*. The elucidation of these mechanisms, through omic approaches (genomics, transcriptomics, proteomics, and metabolomics), may provide insights for the improvement of biological control strategies and plant growth promotion mediated by this microorganism.

Moreover, the discovery of new species of *Streptomyces* sp. (RN 24) and *Kitasatospora* sp. (SOL 195) opens up perspectives for the bioprospecting of bioactive secondary metabolites from these actinomycetes. Future studies may focus on the isolation, structural characterization, and evaluation of the biological activities of the compounds produced by these new species, aiming at the identification of new antimicrobial, antitumor, and immunosuppressive agents, among others.

## 5. Conclusions

The present study revealed the biotechnological potential of bacteria isolated from sediments of the Amazonian rivers Negro and Solimões as biocontrol agents against *R. solanacearum* and plant growth promoters. Among the evaluated isolates, *P. aryabhattai* RN 11 stood out for its efficacy in suppressing the phytopathogen in the soil, reducing the incidence of bacterial wilt, and promoting the growth of tomato plants under different climatic conditions. These results open up promising perspectives for the development of microbial inoculants based on *P. aryabhattai* RN 11, aiming at the integrated management of bacterial wilt in tomato.

Furthermore, the discovery of possible new species of *Streptomyces* (RN 24) and *Kitasatospora* (SOL 195) highlights the importance of the microbial biodiversity of Amazonian rivers as a source of new bioactive compounds and biocontrol agents. These findings emphasize the need for future studies to explore the biotechnological potential of these new species, as well as to investigate the efficacy of the isolates under field conditions and elucidate the molecular mechanisms involved in the plant–microorganism interaction.

## Figures and Tables

**Figure 1 microorganisms-12-01364-f001:**
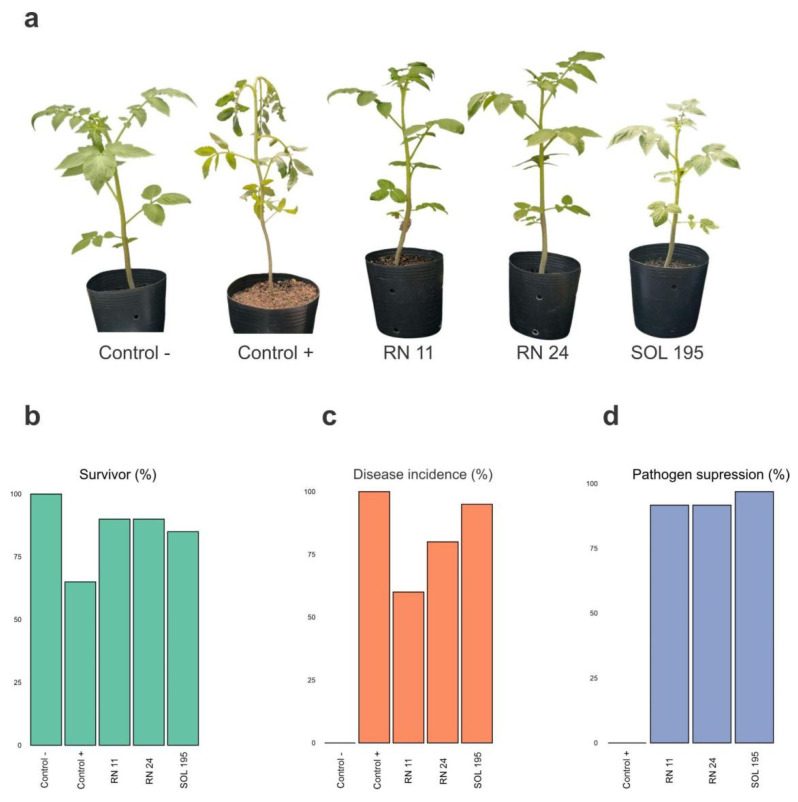
(**a**) Appearance of seedlings from control groups and those treated with *Priestia aryabhattai* RN 11, *Streptomyces* sp. RN 24, and *Kitasatospora* sp. SOL 195. (**b**) Survival indices, (**c**) disease incidence, and (**d**) suppression of *Ralstonia solanacearum* in the soil using the microbial agents as biological controllers during the Amazonian dry period.

**Table 1 microorganisms-12-01364-t001:** Taxonomic identification of bacterial isolates tested in planta for biocontrol potential against *Ralstonia solanacearum*.

Isolate	Size (pb)	Scaffolds	Type Species	NCBI Accession	Specie	ANI (%)	dDDH2 (%)
RN 11	5.262.007	45	*Priestia aryabhattai*	NZ_CP024035	*Priestia aryabhattai*	98.61	88.3
RN 24	8.364.889	366	*Streptomyces ardesiascus*	BEWC01000001.1	*Streptomyces* sp. nov.	92.29	46.8
SOL 195	9.091.611	397	*Kitasatospora aureofaciens*	CP020567.1	*Kitasatospora* sp. nov.	86.36	31.1

**Table 2 microorganisms-12-01364-t002:** Plant growth inducers and enzymes produced in vitro by *Priestia aryabhattai* RN 11, *Streptomyces* sp. RN 24 and *Kitasatospora* sp. SOL 195.

Assay	*P. aryabhattai* RN 11	*Streptomyces* sp. RN 24	*Kitasatospora* sp. SOL 195
Siderophore (mm)	12 ± 1.6	0	5 ± 0.8
IAA (µg/mL)	26.1 ± 2	42.1 ± 1.8	47.8 ± 2.3
Ammonia	+	++	+++
P (mm)	0	0	0
ZnO (mm)	11 ± 1.5	0	0
ZnSO_4_ (mm)	15 ± 2	0	0

+ weak reaction; ++ medium reaction; +++ strong reaction; mean of triplicate.

**Table 3 microorganisms-12-01364-t003:** Growth promotion of tomato cv. San Marzano infected with *Ralstonia solanacearum* using agents *Priestia aryabhattai* RN 11, *Streptomyces* sp. RN 24, and *Kitasatospora* sp. SOL 195 during summer.

Test	Height (cm)	Stem Diameter (cm)	Root (cm)	Branch (unid)	Leaf (cm)	ADW (g)	RDW (g)
NC	51.96 ± 5.9 a	0.40 a	14.77 ± 2.1 a	8 ± 1 b	7.05 ± 0.17 a	0.618 ± 0.06 a	0.116 ± 0.02 c
PC	42.64 ± 7.2 b	0.20 c	10.69 ± 1.7 c	6 ± 1 c	3.77 ± 0.22 c	0.309 ± 0.02 c	0.094 ± 0.01 d
RN 11	46.21 ± 3.1 b	0.36 ± 0.05 a	15.46 ± 2.8 a	8 ± 1 b	7.04 ± 0.12 a	0.525 ± 0.03 b	0.151 ± 0.01 b
RN 24	50.81 ± 3.2 a	0.32 ± 0.07 b	13.37 ± 1.9 b	8 ± 1 b	6.81 ± 0.35 b	0.509 ± 0.02 b	0.142 ± 0.01 b
SOL 195	44.85 ± 3.3 b	0.25 ± 0.05 c	13.96 ± 1.6 b	7 ± 1 b	3.74 ± 0.36 c	0.332 ± 0.01 c	0.116 ± 0.01 c

ADW—air dry weight (total); RDW—root dry weight (total); NC—negative control without pathogen; PC—positive control with pathogen. Means followed by the same letter do not differ by Scott-Knott test at 5% probability.

**Table 4 microorganisms-12-01364-t004:** Growth promotion effect with *Priestia aryabhattai* RN 11 on tomato cv. San Marzano infected with *Ralstonia solanacearum* in the rainy period.

Test	Height (cm)	Stem Diameter (cm)	Root (cm)	Branch (unid)	Leaf (cm)	ADW (g)	RDW (g)
NC	37.69 ± 3.1 b	0.3 ± 0.09 b	27 ± 4.3 a	7 ± 1 b	5.71 ± 0.4 a	1.1 ± 0.28 b	0.266 ± 0.08 a
PC	34.19 ± 2.5 c	0.22 ± 0.02 c	19.73 ± 3.6 b	7 ± 1 b	4.77 ± 0.9 b	0.94 ± 0.11 b	0.126 ± 0.03 b
RN 11	42.04 ± 4.8 a	0.38 ± 0.03 a	29.04 ± 3.6 a	8 ± 1 a	5.77 ± 0.5 a	1.32 ± 0.23 a	0.26 ± 0.07 a

ADW—air dry weight (total); RDW—root dry weight (total); NC—negative control without pathogen; PC—positive control with pathogen. Means followed by the same letter do not differ by Scott-Knott test at 5% probability.

## Data Availability

The raw data supporting the conclusions of this article will be made available by the authors on request.

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
