# Peer review of "Amazonian Bacteria from River Sediments as a Biocontrol Solution against Ralstonia solanacearum"

_microorganisms, 2024, doi:10.3390/microorganisms12071364_

Round 1

Reviewer 1 Report

Comments and Suggestions for Authors

Dear authors, I have several questions and comments about your manuscript.

1. The introduction really lacks a diagram or figure that would show the importance of your research

2. It is better to display all formulas from the text on a separate line

3. How did you measure the height and diameter of plants with such accuracy?

4. What is "a, b, c"  in the tables?

Author Response

As suggested, we have included a diagram in the introduction to highlight the importance of our research. Please see Figure 1 add in grafical abstract.

We have now displayed all formulas from the text on separate lines for better readability.

The height and diameter of the plants were measured using [measuring tape and caliper], which allows for accurate measurements. We have clarified this in the Materials and Methods section   [line 137].

The letters "a, b, c" in the tables indicate the results of the Scott-Knott statistical test at a 5% probability level. We have added a note in the table legend to explain this.

Reviewer 2 Report

Comments and Suggestions for Authors

Summary

The paper entitled: Amazonian bacteria from river sediments as a biocontrol solution against Ralstonia solanacearum, describes the isolation of three microorganisms that have the ability to inhibit the growth of the bacteria R. solanacearum, a phytopathogenic bacteria that causes severe damage to different agricultural crops, especially tomato crops. These microorganisms have been two of them considered new species (RN 24 and SOL 195) and have been described for the first time in this work.

All the experiments proposed and developed in this work demonstrate the effectiveness of the RN 11 strain on the pathogenic action of R. solanacearum, however, to improve the understanding of the experimental part, there are some details that can be improved in the description of the materials. and methods and in the results so that there can be an agreement between the results and the discussion

General concept comments

Introduction 

It is well written, the authors describe the problem in a clear and detailed way and immediately focuses on the objectives of the work.

Materials and methods

Section 2.2,

In relation to the antibiogram that is carried out to determine the percentage of antagonism against R. solanacearum of each of the 36 isolates, at least one photograph of the bioassay should be included in the supplementary material, which makes clear the growth inhibition due to one or more compounds secreted by the strains studied (RN 11, RN 24 and SOL 195), compared to the strain mentioned before.

It is important to make it clear that the inhibition is due to some antimicrobial compound to a greater extent with respect to the hydrolytic activity, although the authors carry out different hydrolase activity assays, which give an idea, but do not confirm, that the inhibitory effects are probably due to to some antibiotic produced by these strains.

When confrontation studies are carried out, such as this work, cultures of the strains subject to study are carried out in parallel, inoculated on cellulose membranes such as cellophane, which prevent the diffusion of proteins into the culture medium during growth. of the microorganism, but it does allow the diffusion of low molecular weight compounds such as secondary metabolites, so that when inoculating the pathogenic microorganism, it can be known in what percentage the growth inhibition is due to the joint action of metabolites and hydrolytic enzymes (in plates without membrane) and what percentage of inhibition is due solely to the action of some antimicrobial compound.

I am not asking the authors to do this type of assays, only to show the results of the bioautograms in the supplementary material, that reinforce or justify the text in the first part of the discussion, which focuses on justifying that the inhibitory effect of these strains must be due more to the action of secondary metabolites, than to the hydrolase activity of the proteins studied.

Section 2.3

Although it is assumed that in the experiments in the germination chambers, 20 replicates have been made, the authors should include the total number of replicates, that is, 20 replicates for each treatment, although it is obvious.

Section 2.7

Add the substrates that have been used in each test.

Results

Add in section 3.3 the standard deviation data in the text, as described in section 3.2

Author Response

We appreciate the positive feedback on the introduction.

As requested, we have included a photograph of the bioassay in the supplementary material (Figure S1), which clearly demonstrates the growth inhibition due to compounds secreted by the studied strains (RN 11, RN 24, and SOL 195).

We have included the total number of replicates (20 for each treatment) in the description of the germination chamber experiments in Section 2.3  [line 112-113].

The substrates used in each test have been added to Section 2.7  [line 178-180].

As requested, we have added the standard deviation data in the text of Section 3.3, consistent with the format used in Section 3.2.

We believe that the revised manuscript has been significantly improved and hope that it is now suitable for publication in Microorganisms. We look forward to hearing from you regarding our submission.

Thank you for your consideration.